# Genome-Wide Investigation and Characterization of SWEET Gene Family with Focus on Their Evolution and Expression during Hormone and Abiotic Stress Response in Maize

**DOI:** 10.3390/genes13101682

**Published:** 2022-09-20

**Authors:** Jialun Zhu, Lu Zhou, Tianfeng Li, Yanye Ruan, Ao Zhang, Xiaomei Dong, Yanshu Zhu, Cong Li, Jinjuan Fan

**Affiliations:** 1College of Bioscience and Biotechnology, Shenyang Agricultural University, Shenyang 110866, China; 2Shenyang Key Laboratory of Maize Genomic Selection Breeding, Shenyang Agricultural University, Shenyang 110866, China

**Keywords:** maize, SWEET gene family, abiotic stress, ABA, expression pattern

## Abstract

The sugar will eventually be exported transporters (SWEET) family is an important group of transport carriers for carbon partitioning in plants and has important functions in growth, development, and abiotic stress tolerance. Although the SWEET family is an important sugar transporter, little is known of the functions of the SWEET family in maize (*Zea mays*), especially in response to abiotic stresses. To further explore the response pattern of maize SWEET to abiotic stress, a bioinformatics-based approach was used to predict and identify the maize *SWEET* gene (*ZmSWEET*) family. Twenty-four *ZmSWEET* genes were identified using the MaizeGDB database. Phylogenetic analysis resolved these twenty-four genes into four clades. One tandem and five segmental duplication events were identified, which played a major role in *ZmSWEET* family expansion. Synteny analysis provided insight into the evolutionary characteristics of the *ZmSWEET* genes with those of three graminaceous crop species. A heatmap showed that most *ZmSWEET* genes responded to at least one type of abiotic stress. By an abscisic acid signaling pathway, among which five genes were significantly induced under NaCl treatment, eight were obviously up-regulated under PEG treatment and five were up-regulated under Cd stress, revealing their potential functions in response to abiotic stress. These findings will help to explain the evolutionary links of the *ZmSWEET* family and contribute to future studies on the functional characteristics of *ZmSWEET* genes, and then improve abiotic stress tolerance in maize through molecular breeding.

## 1. Introduction

Sugar is an important and basic source of carbon to support plant morphogenesis and growth [1]. As a substrate for the production of primary and secondary metabolites, sugar participates in many physiological processes, including energy metabolism and signal transduction, to maintain plant growth and development and respond to stresses [2,3]. Therefore, the transport and distribution of sugar is crucial in plants [4]. Sugar requires carriers to mediate their transport through biofilms, and the transporters include monosaccharide transporters, sucrose transporters, and sugar will eventually be exported transporters (SWEETs) [5]. The SWEET protein family is an important intracellular and intercellular transporter of sugar, and is responsible for the transmembrane transport of sugar for various physiological functions such as providing sucrose for phloem loading, a response to abiotic stress and regulating seed filling, etc. [5,6,7].

SWEET proteins are distinguished by the presence of the MtN3/saliva (MtN3_slv) domain, also termed the PQ-loop repeat. The MtN3_slv domain is made up of three α-helical transmembrane domains (3-TMs). Eukaryotic SWEETs are normally composed of seven TM domains, comprising two tandemly repeated 3-TMs and a single linking TM domain [8]. The MtN3_slv domain was first identified in the *MtN3* nodulation-specific gene of *Medicago truncatula* from constructed cDNA libraries [9]. A homolog of *MtN3* was discovered in the saliva of *Drosophila* as a gene expressed specifically in the salivary glands during embryonic development [10]. Plant SWEET family members can be divided into four groups based on evolutionary relationships [5]. The Clade I and II members have a functional preference for hexose transport, the Clade III members are sucrose transporters, and the Clade IV members are localized to the tonoplast to transport fructose [5,11]. SWEET family genes play a pivotal role in plant growth and development [4,12]. The first identified SWEET gene, named *AtSWEET1*, was reported to have the ability to transport glucose in *Arabidopsis thaliana* [12]. In addition, *SWEET8* and *SWEET13* function in transporting sugar to develop pollen grains in Arabidopsis, and *SWEET9* plays a crucial role as a sucrose transporter for nectar production [5]. *SWEET15* mediates the export of sugar from the endosperm for embryo development [13]. *SWEET17* was the first member of the SWEET gene family identified in plants for the transport of vesicular fructose and was highly expressed in plant roots, playing a specific role in fructose transport across the tonoplast [14]. In rice, the SWEET family genes *OsSWEET11* and *OsSWEET14* are crucial for rice reproductive organ development. The *ossweet11* and *ossweet14* mutants are defective in seed development, exhibiting a small grain and semidwarf phenotype [15]. In maize, *ZmSWEET13* is the major family member responsible for loading sugar in the phloem [16]. In the *zmsweet13a*, *b*, and *c* mutants, the translocation of photosynthetic products is reduced, and soluble sugar and starch accumulate in the leaves, resulting in severe stunting [17]. The *ZmSWEET4c* protein is specifically expressed during seed germination and mediates entry into the scutellum of glucose resulting from starch hydrolysis [18]. *ZmSWEET4* also plays an important role in the development and maturation of maize kernels, and the *zmsweet4c* mutation causes severe crumpling of the kernel, thereby reducing yield [7]. 

Members of the SWEET family are also extensively involved in biotic and abiotic stress responses [19,20]. *Xanthomonas oryzae* pv. *oryzae* (Xoo) infection may induce the expression of a transcription activator-like (TAL) effector, which binds to promoter region-specific regulatory elements of *OsSWEET11*/*14* to activate their expression [21,22]. When the binding site is mutated, the infection capacity of Xoo is lost [22,23]. Abiotic stresses seriously influence the plant growth and development and photosynthetic product transport [14]. Under abiotic stresses, plants monitor carbohydrate redistribution by regulating SWEET expression [14]. Overexpression of *AtSWEET16* and *AtSWEET17* in Arabidopsisimproved the cold tolerance of transgenic plants [24]. Transgenic *Arabidopsis* overexpressing *AtSWEET15* was more sensitive to salt stress and the *atsweet15* mutant shows greater salt tolerance [25]. Heat stress negatively affected the SWEET function of loading, unloading, and long-range transport of sugar in the phloem, ultimately leading to abnormal growth and development of the plant [26]. Under drought stress, *AtSWEET11* and *AtSWEET12* expression was induced and the transport of sucrose from the leaves to the roots was enhanced [27]. Under Cd stress, *PvSWEET24* was significantly induced, while the expressions of *PvSWEET5* and *PvSWEET20* were severely inhibited in the common bean (*Phaseolus vulgaris* L.) [28]. In addition, *ZmSWEET13a*/*13b*/*13c* were sucrose transporters responsive to abiotic stresses and their transcription levels were enhanced by heavy metal cadmium stress in maize [16]. SWEET genes have been shown to be involved in the response to abiotic stresses in other plants, such as rice, watermelon and pomegranate [11,23,29,30].

Abscisic acid (ABA) is a plant hormone widely involved in plant response to abiotic stresses [31,32]. Exogenous ABA treatment may increase the accumulation of sugar and proline in plants and thus enhance the resistance of plants to drought and cold [32]. In addition, ABA affects the sucrose distribution under abiotic stress by influencing the expression of genes involved in sugar synthesis, metabolism, and transport [33]. The expressions of *OsSWEET13* and *OsSWEET15* were increased under the regulation of the ABA-responsive transcription factor *OsbZIP72*, thereby affecting sugar transport in rice for maintenance of homeostasis under abiotic stresses [34].

Maize is one of the most important crops in the world. In addition to being used as food, maize also plays important roles in feed, industry, medical treatment and other aspects [35]. Studies of the gene family are important for the analysis of the origin and prediction of genes functions, and then predict the function of genes [5,29,30]. The *SWEET* gene family has been identified in many plants [29,30]. However, the functions of SWEET proteins are currently poorly understood in maize, especially in response to abiotic stresses. In the present study, twenty-four *SWEET* genes were identified by genome-wide analysis in maize and were classified into four clades. Comprehensive analysis of the *SWEET* genes was performed, including chromosomal distribution, phylogenetic relationships, exon–intron structure, motif composition, gene duplication, and collinearity analysis. The expression patterns of the *SWEET* genes were analyzed in response to polyethylene glycol (PEG), salt (NaCl), cadmium (Cd) stresses and exogenous ABA treatment. The results provide insights for functional characterization of the proteins and important information for further study of the regulatory mechanism of SWEET family members under abiotic stresses in maize. 

## 2. Results

### 2.1. Identification, Phylogenetic Analysis, and Chromosomal Localization of Maize SWEET Genes

Maize SWEETs were identified using a hidden Markov model and BLASTP searches. Twenty-four predicted whole-length *ZmSWEET* genes were identified with Pfam 03083 in the maizegenome. In addition to SWEET proteins predicted with a Pfam MtN3_slv, *ZmSWEET11b*/*13a*/*13b*/*13c*/*14a*/*14b*/*15a*/*15b* were predicted with a Pfam PQ-loop (PF04193). The MaizeGDB website currently lists and names 23 SWEET genes; *Zm00001d043735* on the NCBI website was named *bidirectional sugar transporter SWEET2*, and thus was named *ZmSWEETb-2* in this study for convenience. The MW of the ZmSWEET proteins ranged from 24.71 kDa (*ZmSWEET17a*) to 37.27 kDa (*ZmSWEET14a*), and the proteins comprised 230 (*ZmSWEETb-2*) to 344 (*ZmSWEET14a*) amino acid residues. The pI ranged from 5.1 (*ZmSWEET15b*) to 9.64 (*ZmSWEET13c*) (Table 1). The amino acid sequences for 24 ZmSWEETs, 17 AtSWEETs, and 23 OsSWEETs were aligned, and a phylogenetic analysis performed to investigate evolutionary relationships. According to the evolutionary tree, SWEET genes can be classified into four clades (I to IV). Clade I included five maize SWEET genes (*ZmSWEET1a*/*1b*/*b-2*/*3a*/*3b*), Clade II included five genes (*ZmSWEET4a*/*4b*/*6a*/*6b*/*6c*), Clade III contained 13 genes (*ZmSWEET11a*/*11b*/*12a*/*12b*/*13a*/*13b*/*13c*/*14a*/*14b*/*15a*/*15b*), and Clade IV contained three genes (*ZmSWEET16*/*17a*/*17b*) (Figure 1). Most *ZmSWEET* genes (15 of 24) were located on chromosomes 1, 3, 5, and 8 (Appendix A). Chromosomes 2, 6, 7, and 9 each carried a single *ZmSWEET* gene (Appendix A).

### 2.2. Gene Structure and Conserved Motif Analysis of ZmSWEET Gene Family 

Structural features of the *ZmSWEET* genes were analyzed and conserved motifs were detected using the MEME Suite. The number of exons and introns was similar within a clade. Most of the *ZmSWEET* genes harbored 4–6 exons, but *ZmSWEET3b* had three exons (Figure 2). Similar motif compositions in the same clade were indicative of functional similarity among the members of each clade (Figure 2). The ZmSWEETs in Clades I and II harbored motifs 1, 2, 3, and 5. Clade III ZmSWEETs contained motifs 1, 2, 3, 4, 5, 6, 7, 9, and 10. Cluster IV ZmSWEETs featured motifs 1, 2, 3, 4, 5, and 6. All ZmSWEETs contained motifs 1 and 2, which are typical motifs of the MtN3_slv domain, and each clade shared similar motif features, further supporting the phylogenetic classification of the SWEET family.

### 2.3. Evolutionary Differentiation and Collinearity Analysis of ZmSWEET Genes 

In the present study, only one pair of *ZmSWEET* genes (*ZmSWEET4c/4b*) was identified as a tandem duplication and were located on chromosome 5 (Appendix A). Five pairs of segmental duplication events were identified, comprising *ZmSWEET6a*/*6b*, *ZmSWEET11a*/*11b*, *ZmSWEET12a*/*12b*, *ZmSWEET13b*/*13c*, and *ZmSWEET17a*/*17b* (Figure 3 and Appendix A). In comparison, seven, four, and four pairs of segmental duplication events were identified among SWEET genes in foxtail millet, rice, and sorghum, respectively (Appendix A). 

The syntenic relationships of the ZmSWEET family were analyzed by constructing six intergenomic collinear maps with three dicotyledons (Arabidopsis, *Medicago sativa*, and *Brassica rapa*) and three monocotyledons (*Setaria italica*, *Oryza sativa*, and *Sorghum bicolor*). A syntenic relationship was not detected between the SWEET genes of maize and dicotyledons. In contrast, 17, 15, and 16 pairs of SWEET collinear genes were identified between maize and foxtail millet, rice, and sorghum, respectively (Figure 4 and Appendix A). 

The evolutionary constraints acting on the duplicated genes were estimated by calculating the rates of synonymous substitution (*K*_s_) and non-synonymous substitution (*K*_a_) and *K*_a_/*K*_s_ ratio for duplicated gene pairs in maize, foxtail millet, rice, and sorghum (Appendix A). Values of *K*_a_/*K*_s_ = 1 indicate neutral selection, *K*_a_/*K*_s_ < 1 indicates purifying selection, and *K*_a_/*K*_s_ > 1 indicates positive selection to accelerate evolution. In the four graminaceous crops, *KQL05781* and *KQL14056* were the only gene pair detected with *K*_a_/*K*_s_ close to 1. Other *SWEET* gene pairs had undergone purifying selection and the estimated divergence time ranged from 16 to 86 Mya (Appendix A). 

### 2.4. Promoter Region Analysis of ZmSWEET Gene Family

To analyze the transcriptional regulatory mechanism of the *ZmSWEET* genes, the *cis*-acting elements in the promoter region of the *ZmSWEET* genes were identified using the PlantCARE database. The *cis*-acting elements were identified and classified into three categories, comprising abiotic and biotic stresses, plant hormone response, and plant growth and development (Figure 5 and Appendix A). Among the abiotic and biotic stress elements, two general stress response motifs were detected, namely MYB (CCAAT box) and MYC (CAC ATG box) binding sites, accounting for 29% and 25% of the total number of *cis*-acting elements identified, respectively. Other stress-specific *cis*-acting elements identified were responsive to injury and pathogens, including W-box, TC-rich repeats, and WUN-motif. Among the plant hormone response elements, the ABRE element involved in ABA response was identified in the promoter regions of 21 ZmSWEET members, the TCA-element involved in salicylic acid response was detected in six ZmSWEETs, and the TGACG-motif involved in methyl jasmonate response was identified in 17 ZmSWEETs. Other elements, such as as-1, which is involved in salicylic acid and oxidative stress response, the CGTCA-motif, which is involved in response to methyl jasmonate, and EREs, which are involved in ethylene reactions, accounted for 18%, 18%, and 4%, respectively, of the total number of *cis*-acting elements in this category. Twelve *cis*-acting elements associated with plant growth and development were identified in the *ZmSWEET* gene promoter regions (Figure 5). These results suggested that the *ZmSWEET* genes might be broadly involved in growth and development, and response to various abiotic stresses in maize. 

### 2.5. Expression Profiles of ZmSWEET Genes in Different Tissues and Expression Analysis of ZmSWEET Genes in Response to Abiotic Stresses and Exogenous ABA

To explore the functions of *ZmSWEET* genes, transcriptome analysis was performed to investigate the expression of *ZmSWEET* genes in different tissues at different developmental stages of maize (Figure 6). *ZmSWEET1a*/*4b*/*15b*/*4c*/*15a* were expressed in almost all tissues analyzed. *ZmSWEET4a*/*13a*/*3a*/*13b*/*1b*/*13c*/*17b* were highly expressed in vegetative organs. *ZmSWEET14a*/*14b*/*16/11a*/*11b* were significantly expressed in the seed. *ZmSWEET6b*/*12b* were weakly expressed in all tissues. These findings suggested that the *ZmSWEET* genes play diverse roles in the growth and development of maize. 

To investigate *ZmSWEET* gene expression in response to abiotic stresses, the transcript abundance of the *ZmSWEET* genes was analyzed by qRT-PCR (Figure 7). Under salt stress, the transcription of five *ZmSWEET* genes (*ZmSWEET1a*/*4a*/*6a*/*6b*/*13a*) was significantly induced, whereas seven genes (*ZmSWEET4b*/*11b*/*12a*/*14a*/*14b*/*15a*/*15b*) were significantly inhibited. Under drought stress, the transcription of *ZmSWEET4c*/*14b*/*15a* was significantly increased. Under Cd stress, the transcript abundance of *ZmSWEET1a* was initially increased and thereafter decreased, and the transcript abundance of *ZmSWEET3b*/*11a*/*13b*/*17a* was distinctly increased. Abscisic acid is an important phytohormone involved in plant response to abiotic stresses. Under exogenous ABA treatment, the transcription of 16 *ZmSWEET* genes was induced, of which five genes (*ZmSWEET1a*/*3b*/*12a*/*14a*/*15b*) were significantly induced. Overall, the expression of 23 *ZmSWEET* genes showed different alterations under the abiotic stresses and exogenous ABA treatment, and some genes were affected on multiple treatments (Appendix A). 

## 3. Discussion

Crop growth and development are hampered by abiotic stresses, which affect crop yields [36,37]. Sugar metabolism is an important physiological activity and is involved in the plant response to diverse environmental factors [3]. SWEET proteins are among the carriers of sugars that influence sugar transport and distribution in the plant [5]. In the current study, 24 *ZmSWEET* genes were identified in the maize genome, and one new gene (*ZmSWEETb-2*) was identified on chromosome 3 from the results of a previous study [18]. However, *Zm00001d009365* was not identified in the present study, probably because of the use of different versions of maize genomic data. In addition, 17, 21, 23, and 24 SWEETs were identified in Arabidopsis, rice, sorghum, and foxtail millet, respectively, which was consistent with the results of previous studies [12,29,36]. All evidence suggested that graminaceous species contain a similar number of *SWEET* genes.

The present phylogenetic analysis revealed that SWEET family genes (comprising ZmSWEETs, AtSWEETs, and OsSWEETs) could be divided into four clades (Figure 1). The SWEET proteins from maize and rice tended to cluster together, whereas proteins from Arabidopsis clustered separately in the phylogenetic tree, suggesting that *SWEET* genes may have mainly evolved after monocotyledon–dicotyledon divergence and before the differentiation of graminaceous species. The distinct clades within the gene families may adopt distinct roles during evolution to maximize the flexibility of environmental adaptation [5]. The SWEET proteins of Clades I and II preferentially transport hexoses (such as glucose and fructose), those of Clade III are effective sucrose transporters, according to studies of Arabidopsis and rice, and the proteins of Clade IV are localized to the vesicular membrane and regulate fructose inflow and efflux [5]. The MtN3_slv domain is the signature domain of the SWEET family of proteins and is represented by two predicted motifs, motif 1 and motif 2. The ZmSWEET family members frequently contained motifs 1, 2, 3, 4, and 5, indicating that these motifs are conserved within the ZmSWEET family and may be associated with the basic function of the protein. Motif 6 was detected in Clades III and IV, motifs 7, 9, and 10 were present in Clade III, whereas motif 8 was detected only in Clade II. The motif analysis showed that conserved domains, such as MtN3_slv, were widely present in all members, and unique motifs were detected in each clade (Figure 2). These results implied that, while SWEET genes transport sugar, the substrates may differ between each clade.

The evolution of gene function in higher plants suggests that each isoform is physiologically differentiated with regard to expression sites and regulatory modalities, thus helping the organism to adapt to different environments. SemiSWEETs, the bacterial homologs of SWEETs, are among the smallest transporter proteins known. In both SWEETs and SemiSWEETs, a collection of three TMHs combine to produce a single MtN3 unit [38,39], and the doubling of TMHs enhances the ability of the SWEET protein to transport sucrose. Currently, there are two hypothesized mechanisms to explain the evolution of SWEETs. The first hypothesis suggests that SWEET proteins arose through the replication and fusion of hemiglycan chains (containing three TMHs) [38]. The second hypothesis suggests that SWEET proteins were derived from the fusion of archaeal and bacterial hemiglycan chains [40]. Internal replication of 3-TMH genes must have occurred early in evolution to produce new proteins with 7-TMHs able to translocate greater amounts of sucrose [38]. Therefore, the current *ZmSWEETs* in the maize genome with 6–7 TMHs imply that the replication and fusion of SWEETs may have occurred in the early genome.

For the gene family expansion, three predominant mechanisms are whole-genome duplication (WGD) events, tandem duplication (TD), and chromosomal fragment duplication (SD) [41]. Among these mechanisms, WGD is an important mechanism that provides potential for the evolution of novel gene functions to enable new adaptations for physiological processes, such as plant tolerance of biotic and abiotic stresses, and generation of gene functional diversity, which in turn affects the evolutionary process of species. As a graminaceous crop, maize has undergone four WGD events (i.e., τ-, σ-, ρ-, and mWGD) of which ρ-WGD was graminaceous-specific and occurred ~95–115 Mya [42]. Maize underwent an additional species-specific WGD event (mWGD, ~26 Mya) [42]. In addition to WGD events, tandem and segmental duplications have played a critical role in the expansion of gene families [43]. In the present study, several gene duplication events were identified, including one tandem duplication and five segmental duplication events. The segmental duplication gene pairs *ZmSWEET6a*/*6b*, *ZmSWEET11a*/*11b*, and *ZmSWEET13b*/*13c* contained similar motif architectures. The tandem repeat gene pair *ZmSWEET4b*/*4c* differed somewhat with *ZmSWEET4c* missing motif 3. The motif structures of the remaining two segmental duplication gene pairs (*ZmSWEET12a*/*12b* and *ZmSWEET17a*/*17b*) differed slightly from one another, suggesting that the motif structures may have changed during gene duplication. SWEET genes with different structural domains may have diverged in function during replication. Thus, tandem duplication and segmental duplication are the major forms of SWEET family expansion in maize. The *K*_a_/*K*_s_ ratio of 10 pairs of *ZmSWEET* genes indicated that all had undergone purifying selection. In addition, 17, 15, and 16 maize genes showed collinearity with the foxtail millet, rice, and sorghum genomes, respectively. In contrast, no gene collinearity was observed between maize and Arabidopsis. The results of collinearity analysis suggested that the expansion of the SWEET gene family may have mainly occurred after the divergence of monocotyledons and dicotyledons, and before the divergence of graminaceous species. This conclusion is consistent with previous evolutionary analyses.

The expression data showed different expression patterns of *ZmSWEET* genes in different tissues. *ZmSWEET1a*/*4b*/*4c*/*15a*/*15b* were significantly expressed in almost all tissues analyzed, *ZmSWEET1b*/*3a*/*4a*/*13a*/*13b*/*13c* were substantially expressed in vegetative organs, and *ZmSWEET11a*/*11b*/*14a*/*14b*/*16* were mainly expressed in the endosperm of the seed, indicating that the genes play important roles in sugar transport in different maize tissues (Figure 6). The changes in expression during organ development also suggested that different *ZmSWEET* genes function at different stages of organ development. The expression of *ZmSWEET* genes under several abiotic stresses was analyzed by qRT-PCR. Some *ZmSWEET* genes were significantly induced by high salinity, PEG, or Cd stress. The results implied that certain *ZmSWEET* genes were induced to modify sugar distribution in response to abiotic stresses in maize. A number of *ZmSWEET* genes were induced by exogenous ABA treatment and abiotic stress, such as *ZmSWEET1a*/*4c*/*14b*/*15b*/*16*/*17a*, suggesting that these genes might be under the regulation of the ABA signaling pathway to respond to abiotic stresses. However, several genes were induced by abiotic stress but not by exogenous ABA treatment, such as *ZmSWEET4a*/*6a*/*6b*/*15a*, which implied that these genes were independent of the ABA signaling pathway.

Overall, 24 *ZmSWEET* genes have been identified in the maize genome. These genes function as sugar transporters and play essential roles in maize growth and development as well as in response to biotic and abiotic stresses. In this study, we analyzed the evolutionary relationships and expression patterns of the *ZmSWEET* genes. The results provide insight into the potential functions and characteristics of the *ZmSWEET* genes. The information will contribute to future studies on the biological roles of the *ZmSWEET* genes in maize.

## 4. Materials and Methods

### 4.1. Plant Materials and Treatments

The autogamous maize cultivar “inbred line” was used in the study. Seeds were preserved in our laboratory and incubated in the seedling culture room of the Laboratory of Plant Physiology and Germplasm, Shenyang Agricultural University, from August to December 2020. Seeds of uniform size and full grains were selected. The seeds were disinfected with 75% ethanol and washed with distilled water after 1 min to remove the residual ethanol. The cleaned seeds were evenly sown in seedling pots, covered with vermiculite to a depth of 2 cm, and irrigated with distilled water to allow the vermiculite to absorb sufficient water. The seeds germinated after approximately 3 days and the seedling roots protruded from the bottom grid of the pots. Hoagland’s nutrient solution (pH 6.0) was added to the basal tray in which the seedling pots were placed to ensure that the roots could access the nutrient solution. The nutrient solution was replaced every 3 days until the seedlings attained the three-leaf stage [44]. The three-leaf seedlings were then treated with drought, salt, and Cd stresses and exogenous ABA by application of half-strength Hoagland’s nutrient solution supplemented with 20% PEG for drought stress treatment, 200 mol/L NaCl for salt stress treatment, 40 mg/L CdCl_2_ for Cd stress treatment, or 100 µmol/L ABA. The uppermost mature leaves were collected at 0, 6, 12, and 24 h after initiation of the stress treatment, with three biological replicates at each time point. The sampled leaves were snap-frozen in liquid nitrogen and stored in an ultra-low-temperature refrigerator at −80 °C.

### 4.2. Identification and Evolutionary Analysis

Protein sequences of AtSWEETs of Arabidopsis and OsSWEETs of rice were downloaded from Ensembl (https://asia.ensembl.org/index.html accessed on 1 December 2021). The complete amino acid and nucleotide sequences of *Zea mays* B73 RefGen_v4 were downloaded from MaizeGDB (https://maizegdb.org/ accessed on 1 December 2021). In addition, transcriptomic data were downloaded from MaizeGDB (https://maizegdb.org/ accessed on 27 July 2022). The RNA-seq gene map of maize inbred line B73 included 79 different samples [45]. Expression heat maps were performed using TBtools (https://github.com/CJ-Chen/TBtools accessed on 19 March 2022), and transcriptome data were selected from 21 different maize tissues and developmental periods [46]. Row-scale and log-scale normalization calculations and row clustering were performed on the heat maps. Known SWEET protein sequences were used to construct the hidden Markov model archive and to query the maize dataset using HMMER (http://hmmer.org/ accessed on 5 December 2021) [47]. SWEETs from maize were validated by conducting a BLAST search using SWEETs from rice and Arabidopsis as queries. The conserved domains of identified maize SWEETs were predicted with the PFAM (http://pfam.xfam.org/ accessed on 5 December 2021) and CDD (https://www.ncbi.nlm.nih.gov/cdd/ accessed on 5 December 2021) databases [48,49]. Evolutionary trees were constructed (with 1000 bootstrap replicates) for the SWEETs proteins from Arabidopsis, rice, and maize using MEGA 7.0 (https://www.megasoftware.net/ accessed on 19 March 2022) and ClustalW software (https://www.genome.jp/tools-bin/clustalw accessed on 19 March 2022) [50,51]. Chromosomal localization of the maize SWEET genes was performed using a MapChart (http://mg2c.iask.in/mg2c_v2.0/ accessed on 3 April 2022) based on the chromosomal start and termination information downloaded from MaizeGDB (https://maizegdb.org/ accessed on 3 April 2022) [52]. The multiple covariance scanning toolkit (MCScanX, http://chibba.pgml.uga.edu/mcscan2/MCScanX.zip accessed on 5 April 2022) was used to locate tandem duplicated genes [53].

### 4.3. Sequence Analysis

The ExPASy proteomics server (http://web.expasy.org/protparam/ accessed on 5 December 2021) was used to predict the molecular weight (MW) and isoelectric points (pI) of the maize SWEET proteins (Table 1 and Appendix A) [54]. The MEME Suite (http://meme-suite.org/ accessed on 7 December 2021) was employed to identify the conserved protein motifs of ZmSWEETs, which were further annotated with TBtools (https://github.com/CJ-Chen/TBtools accessed on 7 December 2021) [55]. The domain structure comprised ten motifs, with the motif length ranging from 6 to 50 bp (Appendix A). The gene structure was assessed with GSDS (http://gsds.gao-lab.org/ accessed on 10 December 2021) [56]. The *cis*-acting elements in the 1500 bp sequence upstream of the coding sequences were predicted with the PlantCARE database (Appendix A) (http://bioinformatics.psb.ugent.be/webtools/plantcare/html/ accessed on 13 December 2021). Elements (ABRE, DRE, LTRE, ERE, and MBS) associated with abiotic stress response were subjected to further analysis [57,58].

### 4.4. Replication Events and K_a_/K_s_ Analysis of SWEET Genes

Maize *SWEET* genes were examined for relationships within genomes and between single or multiple intergenomic covariates using MCScanX (http://chibba.pgml.uga.edu/mcscan2/MCScanX.zip accessed on 6 May 2022) [59]. Multiple mechanisms may lead to gene family expansion and doubling, including whole-genome duplication or polyploidization, tandem duplication, segmental duplication, transposon-mediated transposon duplication, and retroposition.

To explore the selection pressure on the maize SWEET family, the ratio of *K*_a_ (non-synonymous substitution rate) and *K*_s_ (synonymous substitution rate) was calculated using ClustalW (https://www.genome.jp/tools-bin/clustalw accessed on 7 May 2022). The time of occurrence of segmental duplication events for homologous genes was calculated as
*T* = *K*_s_/2λ × 10^−6^(1)
where λ is the rate of molecular substitution in grasses (6.5 × 10^−9^), and expressed as a million years ago (Mya) [60].

### 4.5. Quantitative Real-Time PCR Analysis

Total RNA isolation and quantitative real-time PCR (qRT-PCR) analysis were performed to analyze the expression of maize genes under salt, drought, and Cd stress and exogenous ABA treatment [61]. A total of 23 maize SWEET homologous genes were used for the analysis. Total RNA from plant leaves was extracted using TRIzol Reagent (CW Biotech) and subjected to DNase I treatment to remove genomic DNA contamination. The RNA concentration was determined by means of a BioDrop ultra-micro ultraviolet nucleic acid assay. First-strand cDNA was synthesized from 1 µg total RNA using the UEIris II RT-PCR system. The qRT-PCR assays were performed using a real-time PCR analyzer (Bio-Rad, Applied Biosystems PCR, SCILOGEX Gradient Thermal Cycler PCR Instrument TC1000-G). Each reaction mixture contained 10 µL of 2× SYBR^®^ *Green Pro Taq* HS Premix, 1.0 µL cDNA sample, 0.4 µL forward primer (final concentration 10 µM), and 0.4 µL reverse primer (final concentration 10 µM) in a final volume of 20 µL. The thermal-cycling protocol was as follows: 95 °C for 5 min, then 45 cycles of 95 °C for 15 s and 60 °C for 1 min. Melting curve analysis was used to verify the specificity of the reaction. Three technical replicates of each cDNA sample were analyzed. The *Zm00001d013367* genes were selected as an internal control to normalize the transcript levels of *ZmSWEET* genes. The relative gene expression levels were calculated using the 2^−∆∆*C*t^ method (Appendix A). The normalized data were processed with TBtools and plotted as a heatmap to visualize the changes in *SWEET* gene expression (https://github.com/CJ-Chen/TBtools accessed on 19 March 2022) [62]. The number of genes with relative gene expression greater than 2 and relative gene expression less than 0.5 under the four treatments were shown by using the online site Venny 2.1.0 (https://bioinfogp.cnb.csic.es/tools/venny/ accessed on 22 August 2022) to create venny diagrams. All primer pairs were designed with Primer (v5.0) software (http://www.broadinstitute.org/ftp/pub/software/Primer5.0/ accessed on 1 December 2021) and are listed in Appendix A.

## Figures and Tables

**Figure 1 genes-13-01682-f001:**
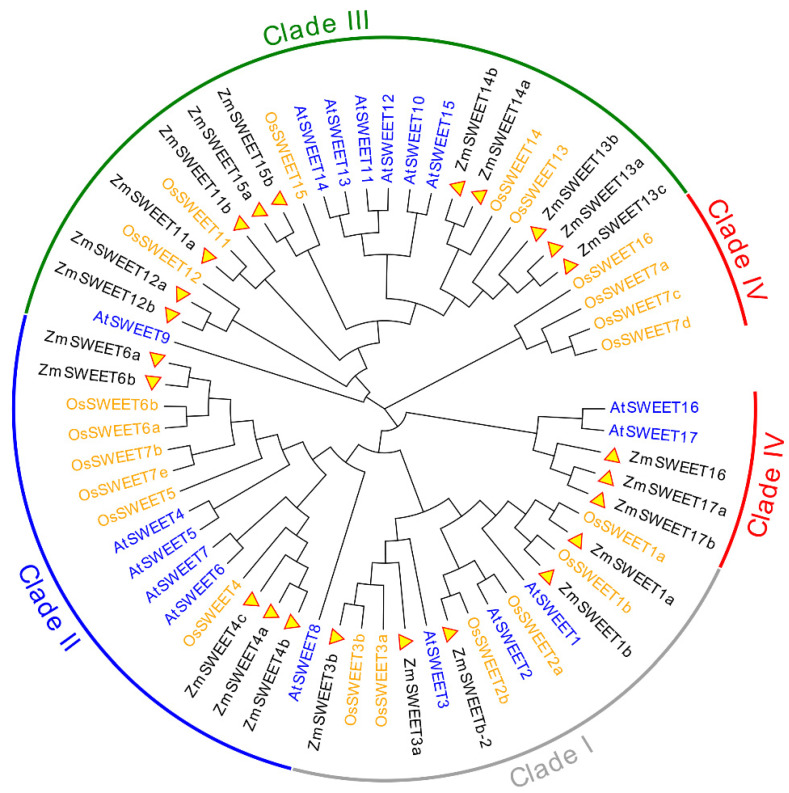
Phylogenetic tree for SWEET proteins of maize, rice, and Arabidopsis. Multiple sequence alignment of the SWEET domains was performed using MUSCLE, and the phylogenetic tree was constructed using MEGA 7.0 with the maximum likelihood method with 1000 bootstrap replicates. The tree was divided into four clades, designated I, II, III, and IV. Proteins of maize, rice, and Arabidopsis are highlighted in black, orange, and blue, respectively.

**Figure 2 genes-13-01682-f002:**
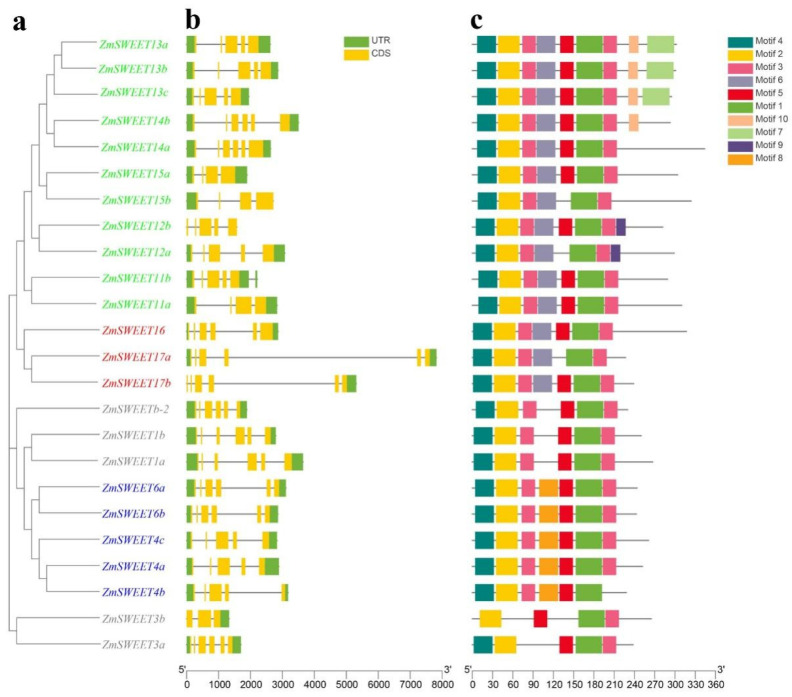
Conserved motifs and gene structure analyses of maize SWEETs based on phylogenetic relationships. All motifs were identified with the MEME Suite using the complete amino acid sequences. Exon–intron structure analyses were performed with TBtools. (**a**) Neighbor-joining tree indicating evolutionary relationships. Clades I, II, III, and IV are indicated in gray, green, blue, and red, respectively. (**b**) Exon–intron structure. Green boxes, yellow boxes, and black lines indicate the untranslated region, coding sequence, and gene length, respectively. (**c**) Conserved motifs.

**Figure 3 genes-13-01682-f003:**
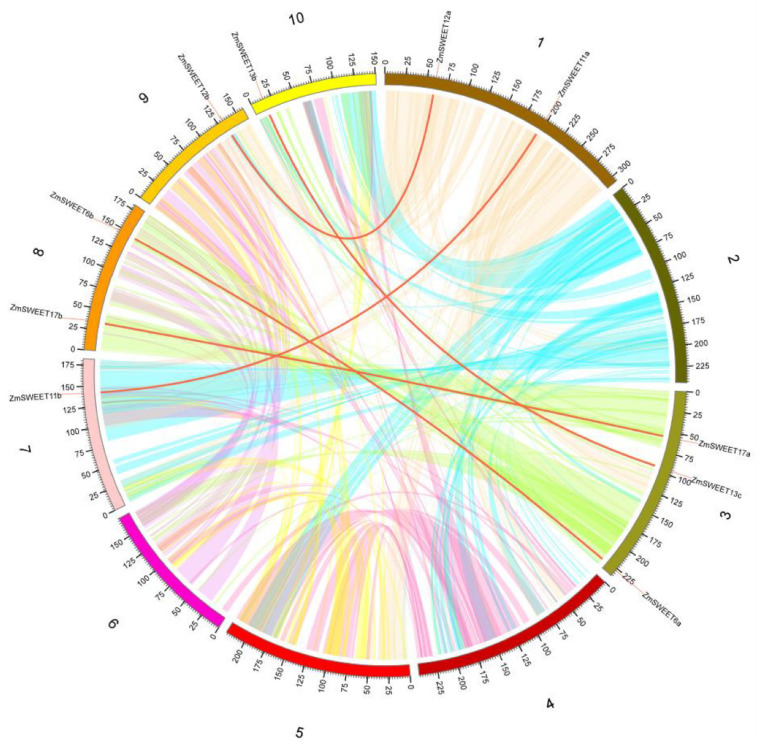
Synteny analysis of the SWEET family in maize. Red curves linking *ZmSWEET* genes indicate duplicated gene pairs in the maize SWEET family.

**Figure 4 genes-13-01682-f004:**
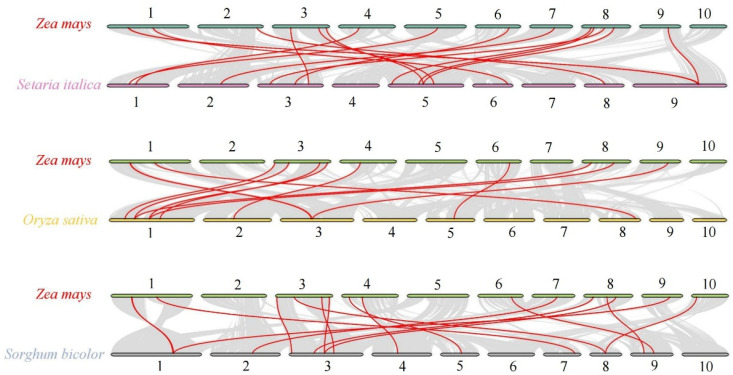
Synteny analysis of SWEET genes between maize and three representative graminaceous species. Gray lines in the background show collinear blocks in the genomes of maize and foxtail millet, rice, and sorghum, and red lines highlight the collinear SWEET gene pairs.

**Figure 5 genes-13-01682-f005:**
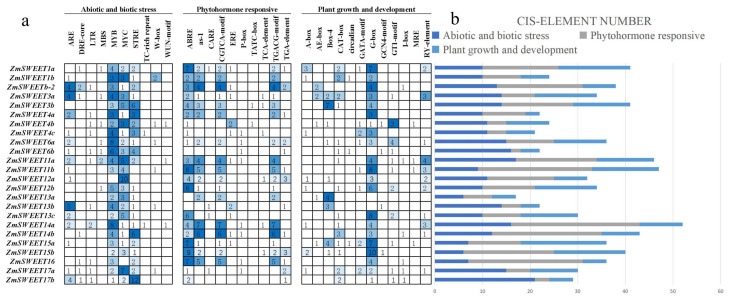
*Cis*-acting elements in the *ZmSWEET* gene family. (**a**) Numbers of different elements in the promoter region of the ZmSWEET genes, as indicated by different color intensities and numbers in the grid. (**b**) Total number of *cis*-acting elements in each response category. Dark blue indicates abiotic and biotic stresses, gray indicates phytohormones, and light blue represents plant growth and development.

**Figure 6 genes-13-01682-f006:**
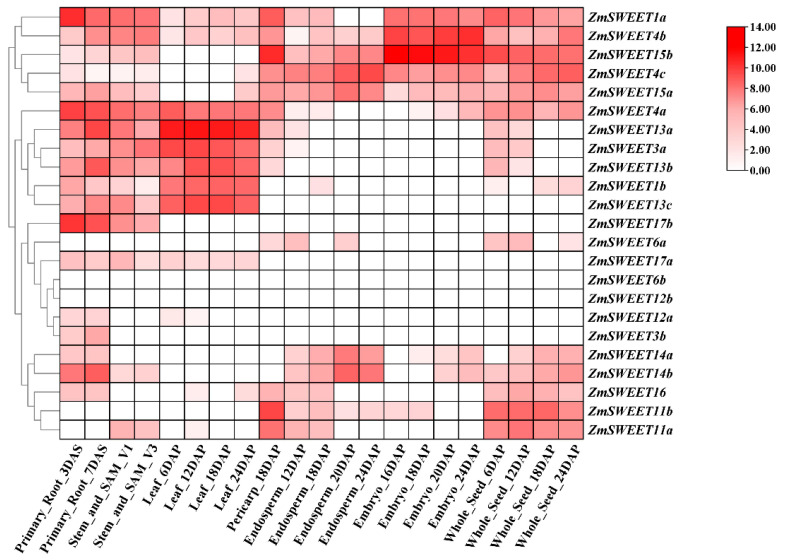
Expression profiles of maize SWEETs in different tissues. Log_2_-based fold change data were used to create the heatmap. Fold changes in gene expression are indicated by the color scale. DAS: Days of growth after sprouting. DAP: Days after pollination.

**Figure 7 genes-13-01682-f007:**
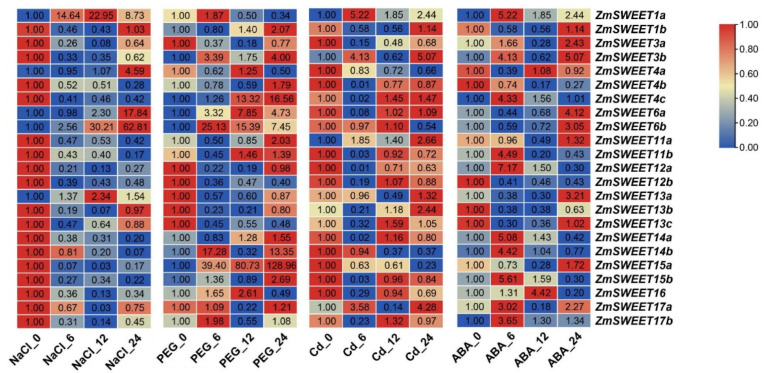
Expression analysis of ZmSWEET in response to abiotic stresses and exogenous ABA. Red represents induced expression; blue represents repressed expression. The 0 h relative expression level of each treatment was taken as 1.

**Table 1 genes-13-01682-t001:** The information of Sugar Will Eventually be Exported Transporter gene family in Maize.

Name	GeneID	Chrom	Transcript	No. of Amino Acids	TMHs	Theoretical pI	Molecular Weight (Average)
ZmSWEET1a	Zm00001d000222	B73V4_ctg26	1	250	6	9.26	28,631.97
ZmSWEET1b	Zm00001d038226	Chrom06	1	267	7	8.9	26,773.47
ZmSWEET3a	Zm00001d010440	Chrom08	2	239	7	9.02	26,104
ZmSWEET3b	Zm00001d039347	Chrom03	1	265	7	9.43	28,672.58
ZmSWEET4a	Zm00001d015905	Chrom05	1	252	6	9.44	27,278.59
ZmSWEET4b	Zm00001d015914	Chrom05	6	255	6	9.48	24,756.56
ZmSWEET4c	Zm00001d015912	Chrom05	1	261	7	9.54	28,279.61
ZmSWEET6a	Zm00001d044421	Chrom03	1	244	6	9.03	27,133.44
ZmSWEET6b	Zm00001d011299	Chrom08	1	243	7	8.87	26,912.16
ZmSWEET11a	Zm00001d031647	Chrom01	1	289	7	9.47	33,490.68
ZmSWEET11b	Zm00001d021064	Chrom07	1	310	7	9.2	31,314.46
ZmSWEET12a	Zm00001d029135	Chrom01	1	306	7	6.82	32,686.56
ZmSWEET12b	Zm00001d047487	Chrom09	1	282	7	9.2	30,837.73
ZmSWEET13a	Zm00001d023677	Chrom10	1	302	7	9.57	32,950.4
ZmSWEET13b	Zm00001d023673	Chrom10	1	301	7	9.57	32,929.51
ZmSWEET13c	Zm00001d041067	Chrom03	1	295	7	9.64	32,280.66
ZmSWEET14a	Zm00001d007365	Chrom02	1	344	7	9.48	37,267.29
ZmSWEET14b	Zm00001d049252	Chrom04	1	293	7	9.28	31,679.58
ZmSWEET15a	Zm00001d050577	Chrom04	1	304	7	5.67	32,944.15
ZmSWEET15b	Zm00001d016590	Chrom05	1	333	6	5.1	34,340.3
ZmSWEET16	Zm00001d029098	Chrom01	1	317	7	6.6	33,998.45
ZmSWEET17a	Zm00001d040656	Chrom03	3	238	6	6.71	17,560.88
ZmSWEET17b	Zm00001d009071	Chrom08	2	239	7	6.1	25,906.62
ZmSWEETb-2	Zm00001d043735	Chrom03	1	230	7	8.75	25,165.85

## Data Availability

Not applicable.

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
