# Peer review of "Genome-Wide Investigation and Characterization of SWEET Gene Family with Focus on Their Evolution and Expression during Hormone and Abiotic Stress Response in Maize"

_genes, 2022, doi:10.3390/genes13101682_

Round 1

Reviewer 1 Report

This article presented Genome-wide investigation and characterization of SWEET gene family with focus on their evolution and expression during hormone and abiotic stress response in maize. The study is well organized and data is well arranged. The findings would be helpful for future studies. There are some shortcomings for that should be resolve.

Overall, the study is well designed and presented in a good way, but mostly the literature is not cited. Grammatical and typos must be revised

Line 22-25 must be revised not clear grammatically.

Add quantitative results of the PEG, salt (NaCl), cadmium (Cd) stresses and exogenous ABA treatment.

The introduction part is well written but still some details are required. The authors should provide details of the maize.

Its economic importance.

Threats of abiotic stresses and its response also link with SWEET gene family.

Line 33 specify the carrier.

Line 37 add names of physiological activities.

The importance of genome wide identification studies.

In introduction also discuss other stresses which are examined in this study.

Line 359 provide the link of GDB.

In discussion line 239 also recommend adding recent reference the following study may be helpful.

https://doi.org/10.3390/ijms22179175,

Section 4.4.  could be cited with https://doi.org/10.1007/s10725-021-00785-7, 

Author Response

Dear reviewer:

We would like to thank you for your comments and suggestions regarding our manuscript entitled “Genome-wide investigation and characterization of SWEET gene family with focus on their evolution and expression during hormone and abiotic stress response in maize” (Manuscript ID: 1910917). We have revised the manuscript to incorporate the suggestions. We have compiled the main corrections and the responses to your comments below. We hope that the corrections will meet your approval.

Responses to the reviewers’ comments:

Reviewer #1:

1- Overall, the study is well designed and presented in a good way, but mostly the literature is not cited. Grammatical and typos must be revised

Response: Thanks for your comment. The manuscript has been checked in full to eliminate grammatical errors and uploaded as a revised manuscript.

2- Line 22-25 must be revised not clear grammatically.

Response: Thanks for your helpful comment. This part has been modified in lines 27 to 30 of revision.

3- Add quantitative results of the PEG, salt (NaCl), cadmium (Cd) stresses and exogenous ABA treatment.

Response: Thank you for your suggestion. The quantitative results of the PEG, salt (NaCl), cadmium (Cd) stresses and exogenous ABA treatmentin have been listed in Table S6 and added to the materials and methods in line 437.

4- The introduction part is well written but still some details are required. The authors should provide details of the maize.

Response: Thank you for your suggestion. Some details in the introduction part have been add to the marked version as follow:

Its economic importance.

Response: Thank you for your suggestion. Maize is one of the most important food crops and has economic value in the world. Descriptions of the economic importance have been added to the introduction part in Lines 108 to 110 of marked version.

Threats of abiotic stresses and its response also link with SWEET gene family.

Response: Thanks for your suggestion. The description about the link of abiotic stresses and SWEET gene family has been modified in lines 83 to 85 in remarked version.

Line 33 specify the carrier.

Response: Thank you for your suggestion. Examples of the carriers have been added in lines 41 to 43 in marked version.

Line 37 add names of physiological activities.

Response: Thank you for your suggestion. ”SWEET is responsible for the transmembrane transport of sugar for various physiological functions such as provide sucrose for phloem loading, response to abiotic stress and regulate seed filling” have been listed in lines 43 to 47.

The importance of genome wide identification studies.

Response: Thanks for your suggestion. The importance of genome wide identification studies has been added in lines 110 to 111 and 114 of marked version.

In introduction also discuss other stresses which are examined in this study.

Response: Thank you for your suggestion. In this study, three abiotic stresses were examined, including drought stress, high salt stress and heavy metal stress. The research progress of different abiotic stresses has been introduced and added to the introduction part in lines 85 to 87 (low temperature stress), 87 to 88 (salt stress), 88 to 91 (heat stress), 91 to 92 (drought stress) and 93 to 97 (heavy metal stress) of the marked version. 

5- Line 359 provide the link of GDB.

Response: We apologize for this omission. The link of GDB has been added in line 379. ”The complete amino acid and nucleotide sequences of Zea mays B73 RefGen_v4 were downloaded from MaizeGDB (https://maizegdb.org/).”

6- In discussion line 239 also recommend adding recent reference the following study may be helpful.

https://doi.org/10.3390/ijms22179175,

Response: Thanks for your suggestion. The reference has been cited in line 259 in revision.

7- Section 4.4.  could be cited with https://doi.org/10.1007/s10725-021-00785-7,

Response: Thanks for your suggestion. The reference has been cited in line 423 in revision.

Reviewer 2 Report

The manuscript entitled “Genome-wide investigation and characterization of SWEET gene family with focus on their evolution and expression during hormone and abiotic stress response in maize” describes bioinformatic analysis of SWEET gene family in corn with the focus on hormonal and abiotic stress response. Currently, several comprehensive studies have performed on SWEET gene family in watermelon, pomegranate, rice and etc. However, as the authors focused on abiotic stress and hormonal responses, this could be the novelty aspect of the present study. Altogether the manuscript needs minor revision to be ready for further consideration. With comments:

1-L15-16: “To explore the mechanism of maize SWEET response to abiotic stress, a bioinformatics-based approach was used to analyze the maize SWEET gene (ZmSWEET) family” could it be achieved without molecular and physiological evidences. We should be more careful about the idea and proposes behind the performed work.

2- The abstract is somewhat null and no data value has been provided. You can some of the most important quantitative data to abstract.  

3- There are some very related works which have not cited in the present study. The:

Xuan, C., Lan, G., Si, F., Zeng, Z., Wang, C., Yadav, V., ... & Zhang, X. (2021). Systematic genome-wide study and expression analysis of SWEET gene family: Sugar transporter family contributes to biotic and abiotic stimuli in watermelon. International journal of molecular sciences22(16), 8407.

Zhang, Xinhui, et al. "Identification, Analysis and Gene Cloning of the SWEET Gene Family Provide Insights into Sugar Transport in Pomegranate (Punica granatum)." International journal of molecular sciences 23.5 (2022): 2471.

Wen, Z., Li, M., Meng, J., Li, P., Cheng, T., Zhang, Q., & Sun, L. (2022). Genome-wide identification of the SWEET gene family mediating the cold stress response in Prunus mume. PeerJ10, e13273.

Du, Y., Li, W., Geng, J., Li, S., Zhang, W., Liu, X., ... & Zhao, Q. (2022). Genome-wide identification of the SWEET gene family in Phaseolus vulgaris L. and their patterns of expression under abiotic stress. Journal of Plant Interactions17(1), 390-403.

4- Please provide all exploited information in material and methods even default parameters of used software. This could include Gene ID of used sequence, gap penalty and gap extension used, exact version of software (in case of on-line tools mention access date) and other cases.

5- The manuscript needs a moderate language polishing which could be made as final check. See two examples:

- Although the SWEET family are important → Although the SWEET family is important

- Sugars are an important and basic source → Sugars are an important and basic sources

Author Response

Dear reviewer:

We would like to thank you for your comments and suggestions regarding our manuscript entitled “Genome-wide investigation and characterization of SWEET gene family with focus on their evolution and expression during hormone and abiotic stress response in maize” (Manuscript ID: 1910917). We have revised the manuscript to incorporate the suggestions. We have compiled the main corrections and the responses to your comments below. We hope that the corrections will meet your approval.

Responses to the reviewers’ comments:

Reviewer #2:

The manuscript entitled “Genome-wide investigation and characterization of SWEET gene family with focus on their evolution and expression during hormone and abiotic stress response in maize” describes bioinformatic analysis of SWEET gene family in corn with the focus on hormonal and abiotic stress response. Currently, several comprehensive studies have performed on SWEET gene family in watermelon, pomegranate, rice and etc. However, as the authors focused on abiotic stress and hormonal responses, this could be the novelty aspect of the present study. Altogether the manuscript needs minor revision to be ready for further consideration. With comments:

1-L15-16: “To explore the mechanism of maize SWEET response to abiotic stress, a bioinformatics-based approach was used to predict the maize SWEET gene (ZmSWEET) family” could it be achieved without molecular and physiological evidences. We should be more careful about the idea and proposes behind the performed work.

Response: Thank you for your valuable suggestion. We have modified the description in lines 17 to 19 to “To further explore the response pattern of maize SWEET to abiotic stress, a bioinformatics-based approach was used to predict and identified the maize SWEET gene (ZmSWEET) family.”

2-The abstract is somewhat null and no data value has been provided. You can some of the most important quantitative data to abstract. 

Response: Thanks for your suggestion. The most important quantitative data has been added to abstract in lines 25 to 27.

3- There are some very related works which have not cited in the present study. The:

Xuan, C., Lan, G., Si, F., Zeng, Z., Wang, C., Yadav, V., ... & Zhang, X. (2021). Systematic genome-wide study and expression analysis of SWEET gene family: Sugar transporter family contributes to biotic and abiotic stimuli in watermelon. International journal of molecular sciences, 22(16), 8407.

Zhang, Xinhui, et al. "Identification, Analysis and Gene Cloning of the SWEET Gene Family Provide Insights into Sugar Transport in Pomegranate (Punica granatum)." International journal of molecular sciences 23.5 (2022): 2471.

Wen, Z., Li, M., Meng, J., Li, P., Cheng, T., Zhang, Q., & Sun, L. (2022). Genome-wide identification of the SWEET gene family mediating the cold stress response in Prunus mume. PeerJ, 10, e13273.

Du, Y., Li, W., Geng, J., Li, S., Zhang, W., Liu, X., ... & Zhao, Q. (2022). Genome-wide identification of the SWEET gene family in Phaseolus vulgaris L. and their patterns of expression under abiotic stress. Journal of Plant Interactions, 17(1), 390-403.

Response: Thank you for the helpful suggestion, Xuan et al., 2021, Zhang et al., 2022 and Du et al., 2022 have been cited in lines 97 to 99, 110 to 112 and 93 to 95 in revision. 

4- Please provide all exploited information in material and methods even default parameters of used software. This could include Gene ID of used sequence, gap penalty and gap extension used, exact version of software (in case of on-line tools mention access date) and other cases.

Response: Thanks for your suggestion. The links of the used websites have been added in revision, including Ensembel, MaizeGDB, HMMER, PFAM, NCBI CDD, ClustalW, MCScanX, ExPASy, MEME, GSDS, PlantCARE, Venny 2.1.0. The softwares used include TBtools and MEGA 7.0. Standardized calculations of row scale and Log scale were performed when normalized for the qRT-PCR relative expression level results using TBtools. Phylogenetic tree using MEGA7.0 was used to construct with the following parameters: phylogenetic test was Btootstrap, model selection Poisson model, Pairwise deletion for differential data processing method, and 1000 replicates. We have indicated Gene ID in the Table 1. 

5- The manuscript needs a moderate language polishing which could be made as final check. See two examples:

- Although the SWEET family are important → Although the SWEET family is important

- Sugars are an important and basic source → Sugars are an important and basic sources

Response: We apologize for this omission. The full text of this manuscript has been modified in revision for similar issues arising in lines 15, 36, 87, 89, 113.

Reviewer 3 Report

The study focus on the identification and characterization of the SWEET gene family important sugar transportation and plant development. The study present significant insights on the genome wide identification study. The study will also helpful in genetic engineering and molecular studies. However, there are some shortcomings which must be resolved.

Why the authors selected this gene family as there are many other important gene families.

Use sugar instead of sugars.

Discuss about the clades in the phylogenetic tree based on previous literature. What are the main reasons that the plants lying in the same clade?

 In methodology also add links of the soft wares or databases.

Conclusion is well presented.

Author Response

Dear reviewer:

We would like to thank you for your comments and suggestions regarding our manuscript entitled “Genome-wide investigation and characterization of SWEET gene family with focus on their evolution and expression during hormone and abiotic stress response in maize” (Manuscript ID: 1910917). We have revised the manuscript to incorporate the suggestions. We have compiled the main corrections and the responses to your comments below. We hope that the corrections will meet your approval.

Responses to the reviewers’ comments:

Reviewer #3:

The study focus on the identification and characterization of the SWEET gene family important sugar transportation and plant development. The study present significant insights on the genome wide identification study. The study will also helpful in genetic engineering and molecular studies. However, there are some shortcomings which must be resolved.

1- Why the authors selected this gene family as there are many other important gene families.

Response: The transport and distribution from source to sink organs of sugar, the product of photosynthesis, are the important factors affecting plant growth and development (Zhang et al., 2022; Lemoine et al., 2013). In addition, abiotic stresses seriously influence on the plant growth and development and sugar transport (Guo et al., 2014; Chen et al., 2014). Members of the SWEET family regulate the transport of different sugars across the cell membrane and control the distribution of sugars in plants (Zhang et al., 2022). Under abiotic stress, plants could redistribute sugar to response to stress by regulating the expression of SWEETs (Guo et al., 2014). Overall, SWEET not only plays a transporter role in sugar transport and distribution, but also may be important for sugar redistribution in response to abiotic stress. Therefore, we investigated the SWEET gene family in this work.

Refence:

Lemoine, R.; La Camera, S.; Atanassova, R.; Dédaldéchamp, F.; Allario, T.; Pourtau, N.; Bonnemain, J.-L.; Laloi, M.; CoutosThévenot, P .; Maurousset, L.; et al. Source-to-sink transport of sugar and regulation by environmental factors. Front. Plant Sci. 2013, 4, 272. [CrossRef]

Zhang, X.H.; Wang, S.; Ren, Y.; Gan, C.Y.; Li, B.B.; Fan, Y.Y.W.; Zhao, X.Q.; Yuan, Z.H. Identification, Analysis and Gene Cloning of the SWEET Gene Family Provide Insights into Sugar Transport in Pomegranate (Punica granatum). International Journal of Molecular Sciences 2022, 23(5): 2471. [CrossRef]

Guo, W.J.; Nagy, R.;Chen, H.Y.; & Martinoia, E. SWEET17, a facilitative transporter, mediates fructose transport across the tonoplast of Arabidopsis roots and leaves. Plant physiology 2014, 164(2): 777-789. [CrossRef]

Chen, L.Q. SWEET sugar transporters for phloem transport and pathogen nutrition. New Phytologist 2014, 201(4): 1150-1155. [CrossRef]

2- Use sugar instead of sugars.

Response: Thanks for your suggestion. The full text of this manuscript has been modified in revisions for similar issues arising in lines 13, 36, 38, 40 to 41, 43 to 45, 64.

3- Discuss about the clades in the phylogenetic tree based on previous literature. What are the main reasons that the plants lying in the same clade?

Response: Phylogenetic trees are used to represent the affinities among species or genes by using a tree clade graph. Identification of the screening sweet was performed by BLAST. The maximum likelihood method (ML) using MEGA 7.0 software was used to phylogenetic trees. By constructing a phylogenetic tree, the amino acid sequences of SWEET genes under the same clade node are similarity, that may mean having the same function. The function of maize SWEET family genes may be predicted from studies employing published SWEET genes in Arabidopsis and rice.

4- In methodology also add links of the soft wares or databases.

Response: The links of the soft wares and databases have been added to materials and methods in line 376 (Ensembl, https://asia.ensembl.org/index.html ), 377 to 379 and 390 (MaizeGDB, https://maizegdb.org/ ), 380 to 381(HMMER, http://hmmer.org/), 383 (PFAM, http://pfam.xfam.org/), 384 (CDD, https://www.ncbi.nlm.nih.gov/cdd/), 386 (MEGA 7.0, https://www.megasoftware.net/), 386 to 387 and 416 (ClustalW, https://www.genome.jp/tools-bin/clustalw), 388 to 389 (MapChart, http://mg2c.iask.in/mg2c_v2.0/), 391 to 392 and 408 to 409 (MCScanX, http://chibba.pgml.uga.edu/mcscan2/MCScanX.zip), 395 (ExPASy, http://web.expasy.org/protparam/), 397 (MEME Suite, http://meme-suite.org/), 399 and 438 to 439 (TBtools, https://github.com/CJ-Chen/TBtools)), 400 to 401 (GSDS, http://gsds.gao-lab.org/), 402 to 403 (PlantCARE database, http://bioinformatics.psb.ugent.be/webtools/plantcare/html/), 441 to 442 (Venny 2.1.0, https://bioinfogp.cnb.csic.es/tools/venny/) and 443 (Primer (v5.0) software, http://www.broadinstitute.org/ftp/pub/software/Primer5.0/).

5- Conclusion is well presented.

Response: Thanks for your comment.
